# DNA evidence of bowhead whale exploitation by Greenlandic Paleo-Inuit 4,000 years ago

Frederik Valeur Seersholm[1,2,†], Mikkel Winther Pedersen[1], Martin Jensen Søe[1,3], Hussein Shokry[1], Sarah Siu Tze Mak[1], Anthony Ruter[1], Maanasa Raghavan[1,4,†], William Fitzhugh[5], Kurt H. Kjær[1], Eske Willerslev[1,4], Morten Meldgaard[1,6], Christian M.O. Kapel[3] & Anders Johannes Hansen[1]

The demographic history of Greenland is characterized by recurrent migrations and extinctions since the first humans arrived 4,500 years ago. Our current understanding of these extinct cultures relies primarily on preserved fossils found in their archaeological deposits, which hold valuable information on past subsistence practices. However, some exploited taxa, though economically important, comprise only a small fraction of these sub-fossil assemblages. Here we reconstruct a comprehensive record of past subsistence economies in Greenland by sequencing ancient DNA from four well-described midden deposits. Our results confirm that the species found in the fossil record, like harp seal and ringed seal, were a vital part of Inuit subsistence, but also add a new dimension with evidence that caribou, walrus and whale species played a more prominent role for the survival of Paleo-Inuit cultures than previously reported. Most notably, we report evidence of bowhead whale exploitation by the Saqqaq culture 4,000 years ago.

[1] Centre for GeoGenetics, Natural History Museum of Denmark, University of Copenhagen, 1350 Copenhagen, Denmark. [2] Trace and Environmental DNA(TrEnD) Laboratory, Department of the Environment and Agriculture, Curtin University, Perth, Western Australia 6102, Australia. [3] Department of Plant and Environmental Sciences, University of Copenhagen, 1871 Frederiksberg, Denmark. [4] Department of Zoology, University of Cambridge, Cambridge CB2 3EJ, UK. [5] Smithsonian Institution, Washington, D.C. 20013–7012, USA. [6] University of Greenland, Manutooq 1, Nuussuaq 3905, Greenland. † Present address: Trace and Environmental DNA (TrEnD) Laboratory, Department of the Environment and Agriculture, Curtin University, Perth, WA 6102, Australia (F.V.S); Department of Zoology, University of Cambridge, Cambridge CB2 3EJ, UK (M.R). Correspondence and requests for materials should be addressed to F.V.S. (email: frederikseersholm@gmail.com) or to M.W.P. (email: mwpedersen@snm.ku.dk).

The population history of the Eastern Arctic is characterized by recurrent migrations into uninhabited lands often followed by local extinctions. This dynamic has resulted in at least three distinct colonization events in Greenland[1]. The earliest Saqqaq Paleo-Inuit (2500–800 BC) were succeeded by late Paleo-Inuit of the Dorset culture (800 BC–1300 AD), and a Viking (Norse) occupation (985–1450 AD)[2], all of which were replaced by the Neo-Inuit from the Thule culture (1200 AD–now)[3]. The survival and collapse of these ancient cultures was highly contingent on their ability to apply new subsistence strategies as even small climatic changes could have a severe impact on the occurrence, frequency and availability of game animals[4,5]. Indeed, the survival of the Norse through the first 100 years of the Little Ice Age (1300–1850 AD)[6] is attributed to a swift transition to a marine diet[7], while the survival of the Saqqaq and Dorset Paleo-Inuit cultures for almost 4,000 years is believed to reflect their ability to shift to new ecological zones when required[1,8].

Midden deposits from historic and prehistoric cultures of Greenland have been extensively excavated during the past 100 years to study the subsistence patterns of these people[4]. However, the taxonomic resolution from highly fragmented assemblages can be low and remains from closely related species can be difficult to distinguish[9]. Finally, traditional osteological analyses seldom identify remains of other organic tissue, such as microfossils, fat, skin and keratinaceous material. Consequently, it is likely that bone counts significantly underestimate the importance of large mammals such as whales, walrus and caribou in the resource economy of Arctic cultures, as it is possible to exploit the meat and fat of these animals without bringing any bones back to a settlement[3,5,9,10].

To expand the current knowledge of subsistence practices in ancient Greenland, we investigate sedimentary ancient DNA (sedaDNA) from four well-described midden deposits at Fladstrand[11,12], Sandnes[9], Qajaa[8,13] and Qeqertasussuk[5,14]. These four sites are characterized by exceptionally high preservation conditions and cover the entire history of human occupation in Greenland represented by remains from Thule, Norse, Dorset and Saqqaq cultures. We reconstruct the faunal assemblage at each site using a specifically designed computational pipeline to detect and quantify vertebrate traces in the metagenomic DNA data. Furthermore, using the same approach on samples enriched for parasitic eggs, we identify host-specific helminths (parasitic worms) in the sediment. With the relative narrow host range of helminth parasites[15], we correlate specific parasites with specific hosts, which provides an independent confirmation of key vertebrate hosts. As sedaDNA can derive from a wide array of organic sources such as skin, meat, fat, urine, faeces and hair[16], we are able to directly quantify the contributions from all organic matter in the sediment, independent of diagnostic fossils. With this data, we demonstrate that caribou, walrus and whale played a more prominent role in the subsistence of the Paleo-Inuit cultures than previously believed. Most notably, we identify large proportions of bowhead whale DNA in the midden deposits at two Paleo-Inuit sites, Qajaa and Qeqertasussuk, demonstrating the first tangible evidence of extensive bowhead whale exploitation by Saqqaq people 4,000 years ago.

## Results

**Sediment samples and DNA sequencing.** In all, 34 sediment samples were collected from stratigraphic sections in the midden deposits at Qajaa[8,13], Qeqertasussuk[5], Sandnes[9,17] and Fladstrand[11,12,18] (Fig. 1). From these we generated 31 shotgun libraries based on the total DNA extracted from midden sediment (hereafter referred to as sediment libraries) and 27 shotgun libraries based on helminth eggs isolated from the sediment by sieving (hereafter referred to as helminth libraries). After initial bioinformatic pre-processing (see Methods) these yielded a total of 2,064,856,802 DNA reads. For downstream analyses, total and helminth shotgun libraries from the same layers were merged to increase the sample size and ensure sufficient vertebrate read counts for robust and reliable comparison between the strata (Fig. 1).

**Taxonomic identification of vertebrates and helminths.** Taxonomic assignment of vertebrates and parasitic worms based on sequence identity was done by aligning all reads against the NCBI database of full mitochondrial genomes within Metazoa (see Methods). After rigorous filtering (see Methods, Supplementary Note 1), 16,366 mitochondrial reads could be unambiguously assigned to the family level or lower within vertebrates. In total, 42 vertebrate taxa, consisting of 1 subspecies, 23 species, 9 genera, 3 subfamilies and 6 families were identified (Supplementary Tables 1 and 2). Similarly, by filtering all reads assigned within the families Toxocaridae and Taeniidae we obtained a total of 465 reads of which 460 originated from the helminth shotgun libraries. We identified 12 different helminth taxa, of which the two most abundant species of the genus Taenia (Taenia hydatigena and Taenia multiceps) and the most abundant species of the genera Echinococcus and Toxocara (Echinococcus canadensis and Toxocara canis, respectively) were selected for further analysis.

Both the vertebrate and the helminth taxonomic profile (Supplementary Tables 1–3) revealed distinct patterns for each of the four cultures investigated. At Fladstrand, the vertebrate reads were dominated by dog or wolf (Canis lupus; 75.8%), with fewer reads from seals (Pagophilus groenlandicus and Pusa hispida), narwhal (Monodon monoceros), caribou (Rangifer tarandus) and hare (Lepus). A canine tapeworm, T. multiceps, was identified in the same layer consistent with the presence of dog or wolf (definitive host), as well as a caribou population (intermediate host) in the vicinity of the settlement[19] (Supplementary Tables 3 and 4). In agreement with the bone record, a very different faunal profile was identified at the Norse settlement of Sandnes, consisting primarily of the domestic animals: cow (Bos, 55.9–70.7%); sheep (Ovis, 7.2–14.3%); and goat (Capra, 8.2–22.4%), with fewer reads from wild fauna such as seals, caribou and walrus (Odobenus rosmarus). The presence of caribou and domestic livestock at Sandnes were indirectly confirmed by the presence of the canine tapeworms: T. hydatigena[20], with sheep/goat as typical intermediate hosts; and T. multiceps[21] and E. canadensis G10, with wild ungulates such as caribou serving as typical intermediate hosts[19,22]. Finally, we found that marine mammals dominated the faunal profiles from the Paleo-Inuit cultures. While the presumed Dorset layer in Qajaa (Fig. 1) consisted mainly of DNA from ringed seal (P. hispida; 52.4%) and dog (27.6%), the Saqqaq layers at both Qajaa and Qeqertasussuk were characterized by large fractions of bowhead whale (Balaena mysticetus; 5.2–49.2%) and harp seal (P. groenlandicus; 24.2–83.9%) DNA. At Qajaa, the identification of the dog roundworm, T. canis with its direct life cycle in canines, is consistent with the presence of canids at the settlement[8].

Finally, we investigated whether the faunal assemblages mirrored the cultural differences using a non-metric multi-dimensional scaling ordination analysis based on Bray–Curtis similarity measures (Supplementary Fig. 1 and Supplementary Methods). We found a clear separation on the first axis reflecting the difference in faunal subsistence between the Norse and the ancient Inuit cultures. More importantly, the second axis was able to differentiate between the three ancient Inuit cultures while clustering Saqqaq layers from two different sites together.

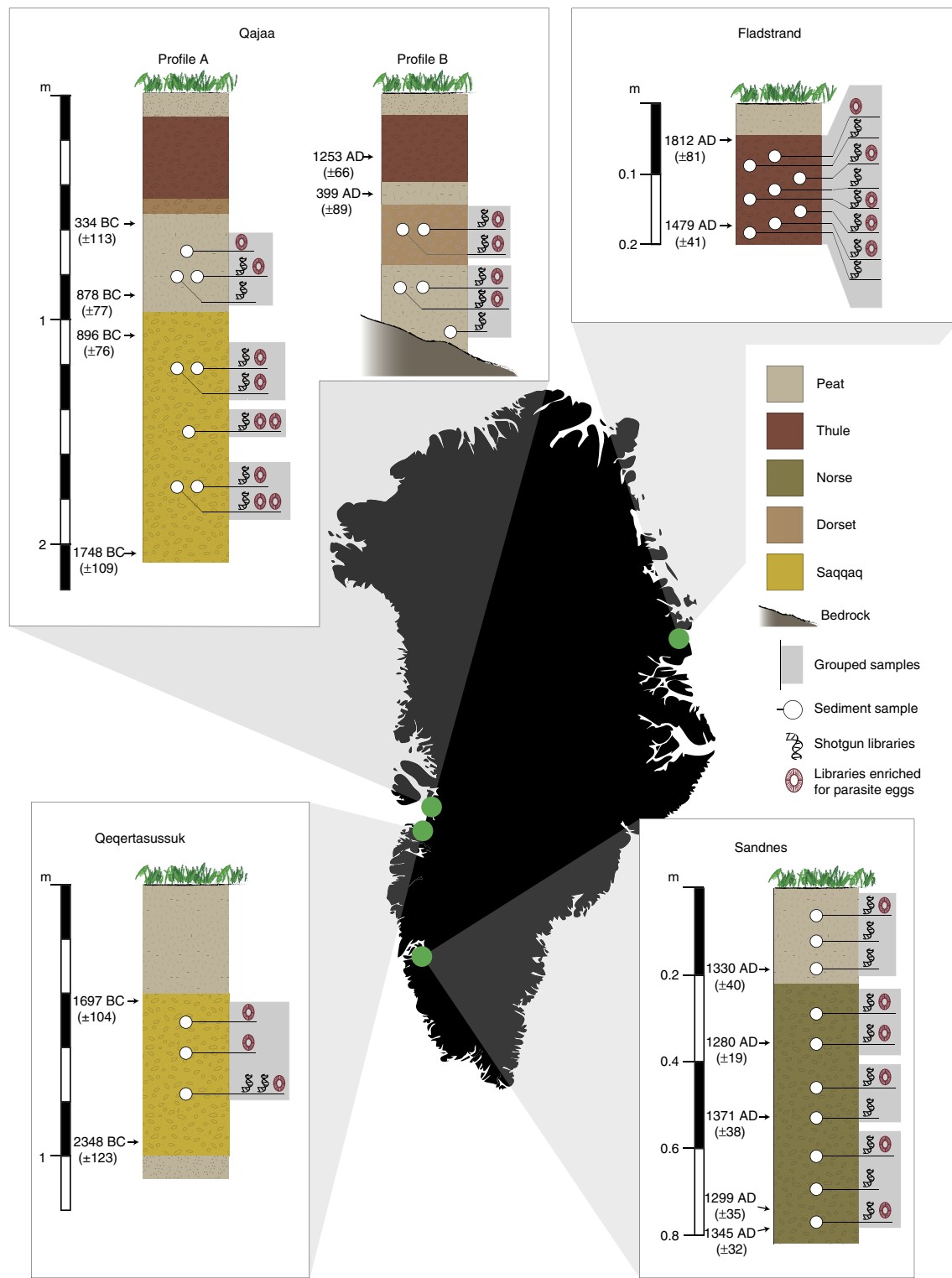

**Figure 1 | Sampling locations and stratigraphic profiles.** The cultural layer of profile B at Qajaa is presumed to be of Dorset origin due to the presence of Dorset microblades. Datings of sedimentary layers are based on the following: Qajaa—Møhl *et al.*[8]; Fladstrand—Gotfredsen *et al.*[11]; Qeqertasussuk—Meldgaard[5]; and Sandnes—this study (Supplementary Table 10).

**Plant DNA**. The plant content of the sediment yielded a total of 351,440 reads when aligned to the complete full plastid database from NCBI (see Methods). Owing to the relatively low number of full chloroplast genomes available (1,006 species), reads were assigned at family level as the lowest taxonomic resolution (Supplementary Table 5). Grasses (*Poaceae*) and willows

(*Salicaceae*) were the most common families identified in all of the anthropogenic layers analysed. Sandnes and Qeqertasussuk were characterized by high concentrations of horsetails (*Equisetaceae*: 36.8–50.2%), while Qajaa had a varied composition across the different sedimentary layers at profiles A and B, constituted by the buttercup family (Ranunculaceae), the heather

family (Ericaceae), sedges (Cyperaceae) and the mosses of Orthotrichaceae. These plant families are consistent with the current vegetation cover in Greenland dominated by grasses (Poaceae) and low-lying shrubs such as dwarf willows (Salicaceae) and crow berries (Ericaceae). This suggests that the plant DNA identified here represents the vegetation cover at the midden for each habitation period rather than plants of significant value to subsistence. To support the metagenomic data, plants at Sandnes were also analysed by *trn*L metabarcoding (Supplementary Fig. 2 and Supplementary Methods). *trn*L metabarcoding results overlapped with the metagenomic results identifying field horsetail (*Equisetum arvense*) as the most abundant plant in all sedimentary layers at Sandnes. Furthermore, the metabarcoding approach detected crowberry (*Empetrum nigrum*) and bentgrasses (Agrostinidae), suggesting that the metagenomic reads assigned to the Ericaceae and Poaceae families represent crowberry and bentgrasses, respectively.

**DNA damage estimation**. We used reads assigned to the 10 most abundant plant families to assess the degree of 5′ C to T misincorporations (aDNA damage)[23] in the samples. DNA damage for vertebrates could not be calculated for the majority of the samples, as read counts from a single species were insufficient for robust DNA damage calculations (<500 reads). However, two samples at Qeqertasussuk and Qajaa contained sufficient reads assigned to bowhead whale, harp seal and caribou to estimate the DNA damage. The youngest samples from Fladstrand, Sandnes and profile B at Qajaa demonstrated no apparent DNA damage (0–2.8%), while the Saqqaq layers of Qajaa and the peat layer from profile A at Qajaa displayed elevated levels of DNA damage (5.0–10.7%). At Qeqertasussuk, a high level of DNA damage was observed for mammal species (7.1–13.1%), while a very low level was observed for the family of horsetails (1.2%). The remaining plant families at Qeqertassusuuk displayed slightly elevated DNA damage signals (3.7–4.8). This suggests that the horsetail DNA at Qeqertasussuk, could represent modern DNA from the long roots of horsetails species growing at the surface[5]. Furthermore, the high level of DNA damage for mammal species at Qajaa (harp seal and caribou) and Qeqertasussuk (harp seal, caribou and bowhead whale) confirms that these signals represent authentic DNA deposited at the time of occupation at the two sites (Supplementary Table 6 and Supplementary Note 2).

**Correlating aDNA analyses with the existing fossil record**. Apart from the presumed Dorset layer at Qajaa, bone fragments recovered from all midden layers in the present study have previously been examined using morphological techniques, thus allowing for a comparative assessment of the aDNA performance similar to comparisons reported in previous studies[24–26]. In total, aDNA identifies 6/11, 9/13, 7/9 and 6/14 of the previously documented species at Fladstrand, Sandnes, Qajaa and Qeqertasussuk, respectively (Fig. 3a), and identifies the presence of previously unreported species such as bowhead whale, walrus and hooded seal (*Cystophora cristata*) at Qajaa and bowhead whale at Qeqertasussuk (Fig. 3a).

Considering the overlap between the species identified by aDNA and zooarchaeology (Fig. 3a), we explored the possibility of estimating the abundance of each species using the DNA read counts. To address this, two libraries from the Saqqaq layers at Qeqertasussuk and Qajaa with high concentrations of vertebrate DNA were chosen for re-sequencing to estimate abundances of each mammalian species with higher confidence. For Qeqertasussuk, we correlated the DNA data with the raw bone count, that is, number of identified specimens (NISP), and calculations of biomass from the most common species identified at Qeqertasussuk[5] (Methods, Supplementary Table 7). We found a strong correlation between the expected biomass and DNA read counts (Pearson's rho = 1.00, P = 0.002), while the correlation between raw bone counts (NISP) and the DNA read counts was less pronounced (Pearson's rho = 0.40, P = 0.6; Fig. 3b). These relationships, in agreement with previous results[27], demonstrate that the DNA read counts reflect the expected biomass estimated from bone counts for individual species rather than the raw bone counts. At Qajaa, the zooarchaeological investigations did not include a calculation of expected biomass for the main species. However, there is a positive correlation between NISP counts and counts of DNA reads at Qajaa (Pearson's rho = 0.93, P = 3.2e-5) and both proxies identify harp seal as the most abundant mammal and seagulls (*Larus*) as the most abundant genus among birds in the Saqqaq layers[8] (Supplementary Table 8). Finally, we found that the species absent in the DNA record for both Qajaa and Qeqertasussuk were species with relatively low biomass and abundance, such as arctic fox, arctic hare and dog.

The correlations between *sed*aDNA read counts and expected biomass raises new questions about Saqqaq subsistence, the most striking being the relative abundance of bowhead whale DNA in the midden deposits of Qajaa and Qeqertasussuk. To provide further support for the identification of bowhead whale at Qeqertasussuk and Qajaa, we realigned all bowhead whale reads from the two re-sequenced libraries against the mitochondrial genome of *B. mysticetus* (gi:38707506), which resulted in assembly of two consensus mitochondrial genomes (1.3 × and 5.8 × coverage depth), from the Qajaa and Qeqertasussuk library, respectively. These mitochondrial genomes displayed distinct ancient DNA damage patterns and clustered together with *B. mysticetus* in a well-supported topology (all relevant posterior probabilities = 1) (Supplementary Fig. 3 and Supplementary Methods). Furthermore, two harp seal mitochondrial genomes (24.1 × and 3.5 × coverage depth) were also recovered from the re-sequenced libraries in addition to a mitochondrial genome from *T. hydatigena* (1.0 × coverage depth) from the bottom layer at the Norse settlement site, Sandnes. As demonstrated in Supplementary Figs 3–5, these mitochondrial genomes display DNA damage comparable to the levels observed in the plant DNA from the same layers (Fig. 2). While these mitogenomes serve as evidence for the identification of harp seal, bowhead whale and *T. hydatigena*, they should not be regarded as sequences from a single individual. Rather, these mitogenomes most likely represent a subset of the individuals present in the sediment layer from which they were retrieved.

## Discussion

Traditionally, midden deposits have been investigated using zooarchaeological approaches, which typically require excavations of large volumes of sediment. Here we used a less intrusive DNA-based alternative that is independent of diagnostic bone fragments and is able to detect remains from a range of different organic sources such as skin, meat, fat, keratinaceous material and bone. Using a metagenomic approach, we characterized the faunal and floral diversity in four ancient midden deposits ranging as far back as ~2000 BC.

Moreover, we were able to recover a diverse faunal profile at Fladstrand despite the high concentration of DNA from dog or wolf at this site. Apart from *C. lupus*, the faunal assemblage constituted hare, seals, caribou and narwhal, all of which are in agreement with the archaeological record[11]. The presence of the dog tapeworm *T. multiceps* suggests that dogs faeces were present in the sampled sediments. These results imply that the sampling site at Fladstrand might have been used as a tethering place for dogs as well as a waste dump. Unfortunately, the dominance of

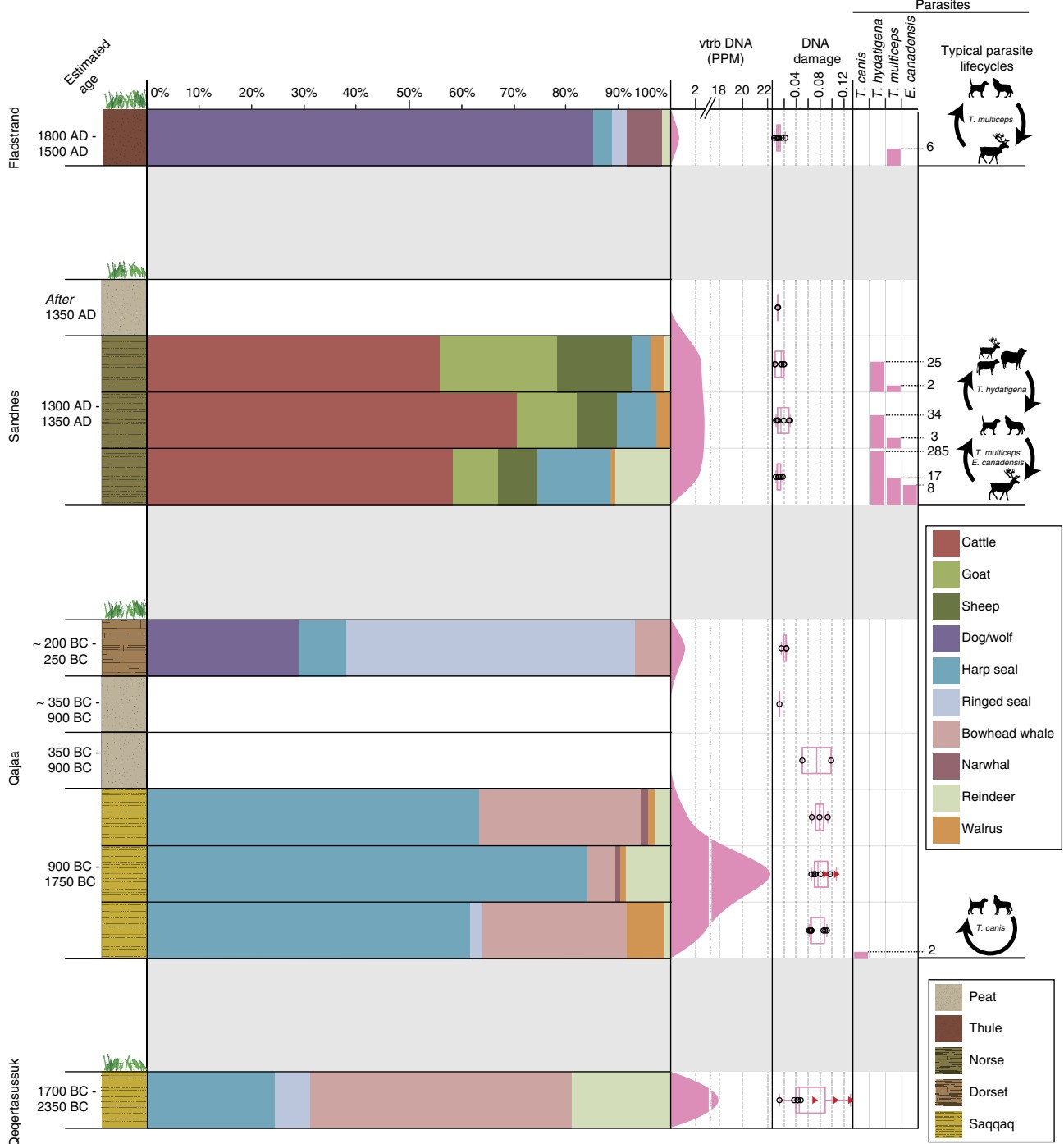

**Figure 2 | 4000 years of resource economy in Greenland.** Barplots represent relative abundances of the 10 most common mammal species identified across all sites (Supplementary Table 1). In cases where higher-order taxa could be uniquely identified to a single species, reads were collapsed to species level (outlined in Methods). Vertebrate DNA concentration is defined as vertebrate DNA reads per million reads analysed (Supplementary Table 12). DNA damage patterns are illustrated by the frequency of C to T transitions in the first 5′ position. Open circles represent DNA damage from plant species while red triangles represents mammal DNA damage. DNA damage calculations are based on species represented by more than 500 reads. Parasite read counts represent DNA reads assigned uniquely to the given parasite (Supplementary Table 3). At Qajaa the two uppermost layers represent profile B, while the four layers below represent profile A. Graphics credits: Dog silhouette by Abujoy/CC BY. Cattle silhouette from clipartkid. Caribou silhouette by mystica/CC.

dog and the low concentration of other vertebrate DNA prevented a meaningful comparison of abundances in this layer with the biomass estimates from zooarchaeological analyses.

We found the midden samples from the Norse settlement of Sandnes to represent a short period in the fourteenth century out of the full settlement history at the site. This explains why the

aDNA record exhibited no marked difference between the anthropogenic layers studied. However, the important status of Sandnes[9,28] was evident in the faunal assemblage, which indicated that the subsistence on the farm was based primarily on domestic cattle, sheep and goat, supplemented with wild fauna such as seals, walrus and caribou. Although the species identified with

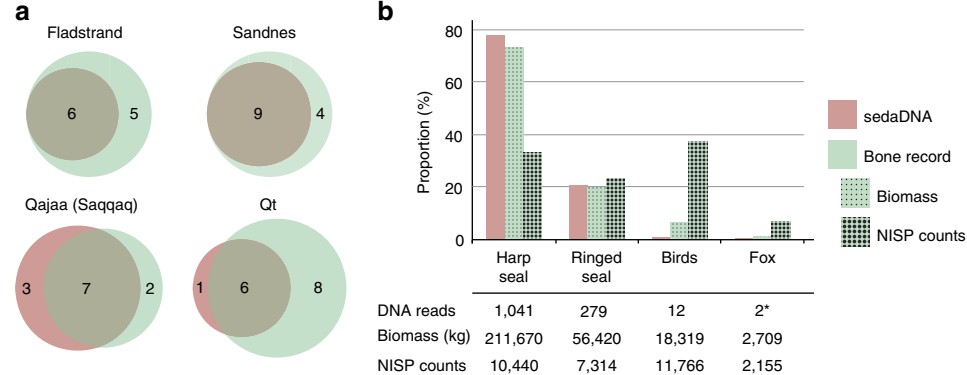

**Figure 3 | Comparison of biodiversity as identified by *sed*aDNA and zooarchaeological excavations.** (**a**) Number of mammals unambiguously identified by *sed*aDNA and in the bone records. (**b**) Relative proportions of the four most common groups identified in the bone record from Qeqertasussuk[5] compared with DNA read counts from Qeqertasussuk (Methods, Supplementary Table 7). *Reads identified, although not included in the quantification analyses due to low read counts.

*sed*aDNA were in agreement with the bone record[9], the biomass estimated from the DNA data found livestock species to be ~3 × higher than the values estimated from the bone record. This is supported by the higher number of uniquely assigned reads to *T. hydatigena*, indicative of a parasitic life cycle typical for domesticated ruminants, namely sheep, compared with less abundant *T. multiceps* and *E. canadensis*, indicative of life cycles in wild life hosts. Like the presence of dog at Fladstrand, this might reflect the presence of livestock species at Sandnes the year round, causing continuous accumulation of DNA in the midden deposits from urine and defecation[29].

The ordination analysis clearly differentiates between the diet based on both domestic and marine animals for the Norse and the extreme reliance on marine ressources for the Inuit (Supplementary Fig. 1). This difference in subsistence practices is also reflected in stable isotope data from human bone remains from Saqqaq[30], Dorset[31], Thule[32] and Norse[32,33] cultural sites. Analysis of isotope composition in Inuit remains suggests a strong dependence on marine resources, while the Norse bone remains show evidence of subsistence based on both domestic and marine animals in comparable quantities.

The ordination analysis also distinguishes the three ancient Inuit cultures from each other, with layers from the same culture at different sites clustering together (Supplementary Fig. 1). In addition, the clear separation of the presumed Dorset layer at Qajaa in the ordination analysis confirms that this layer represents a distinct culture. This is in agreement with the identification of Dorset-like microblades, which further supports that this layer is of Dorset origin. Despite the low concentration of vertebrate DNA in this layer, we identify two novel species for the Dorset culture—bowhead whale and dog. Of the 29 canine reads identified in this layer, 7 could be assigned uniquely to dog (*C. lupus familiaris*), while two reads could be assigned unambiguously to wolf (*C. lupus lupus*). This represents the first identification of dog in the Dorset culture. However, the archaeological context from this layer remains unclear as these samples represent a profile outside of the main excavation area investigated originally by J. Meldgaard[13].

Bowhead whale DNA was identified at both Saqqaq settlements (Fig. 2), in all sediment layers analysed. At Qeqertasussuk, bowhead whale was the most abundant species identified, constituting 49.2% of the DNA, while at Qajaa it ranked as the second or third most abundant species in each layer. These findings are in striking contrast to the bone record. At Qeqertasussuk, only 102 fragments of whale bone, teeth and baleen were found among a total of ~100,000 excavated bones

(0.04%) and, of these, only a single piece of baleen could be identified as either bowhead whale or North Atlantic right whale (*Eubalaena glacialis*)[5]. Similarly, at Qajaa, two narwhal bones were found along with a single bone from an unidentified cetacean out of ~15,000 bones (0.02%)[8]. The underrepresentation of whale bones in archaeological sites is a well-known phenomenon, typically ascribed to difficulties in transporting large carcasses from shore to the settlement[34,35] in combination with the higher value of blubber or meat compared with bones[36]. In the arctic, several studies have suggested that the fossil record may underestimate the importance of whales to ancient Arctic cultures[4,5,9], however, the lack of suitable methods to detect remains of tissue like blubber and meat in sediment have prevented further investigations on this matter. As such, our findings represent the first tangible evidence that bone counts alone may underestimate large whales in Arctic midden remains.

Furthermore, *sed*aDNA results from Qajaa support that the fossil record underestimates other large mammals, such as caribou, walrus and narwhal, all of which have a higher representation in the *sed*aDNA faunal assemblage than observed in the fossil record. Likewise, at Qeqertasussuk, we found caribou to comprise 18.8% of total DNA reads, compared with an estimated 0.3% in the bone record.

In summary, our results demonstrate that large mammals such as caribou, walrus, narwhal and bowhead whale are underrepresented in the osteological record while domestic species such as cow, sheep, goat and dog are overrepresented in the DNA profile. Hence, to confidently reconstruct subsistence practices from midden remains, it is strongly encouraged to apply a combination of *sed*aDNA and morphological analyses, as both of these approaches may be misinterpreted when standing alone (discussed further in Supplementary Note 3).

The identification of bowhead DNA in 4,000-year-old Saqqaq deposits raises questions about the history of whale hunting and whale scavenging centred in the North Pacific and Bering Strait. The origins of active whaling has been tied to the development of toggling harpoons that appear about 4000 BC among North Pacific and Bering Sea peoples for hunting small sea mammals like seals in ice-infested waters[37]. Intermediate-sized toggling harpoons suitable for hunting walrus appear in Alaskan Old Whaling culture ca. 1000 BC (refs 38,39), and large whaling harpoons and floats in Old Bering Sea and Norton cultures between 500 and 800 AD (ref. 40). Systematic whaling with large umiak boat crews became a central economic feature of the Thule culture that migrated into the Eastern Arctic and Greenland around 1200–1400 AD, replacing Dorset Paleo-Inuit culture

whose main quarry were seals, walrus and caribou[41,42]. So far, a single Saqqaq harpoon measuring 16.6 cm remains the only example of large toggling harpoons suitable for hunting large whales in the Paleo-Inuit record[4].

As opposed to whale hunting, scavenging of stranded cetacean carcases was common in pre-historic times and has been described across multiple sites in Europe[43], North America[44] and Africa[34]. Hence, the relative abundance of bowhead whale DNA in the Saqqaq sediment layers could be explained by scavenging whale carcasses. Whales were probably abundant along the nutrient-rich West Greenland waters that were so attractive to European whalers, and dead (drift) whales could have been driven ashore by wind and tides making them available to Saqqaq beach-comers. The warmer Saqqaq climate may also have influenced the frequency of whale strandings; today killer whales appearing in the less ice-congested Arctic waters often cause whales and other sea mammals to seek shelter in shallow bays and inlets, causing them to strand. Dependent on the rate of decomposition, the meat and blubber from drift whales might have been used for human food, oil lamps or feeding of dogs.

On the other hand, large whale hunting is not contingent on Thule style technology. In the Paamiut area in Southern Greenland, humpback whales were traditionally hunted using simple lances and toggling harpoons. By approaching the docile animals noiselessly, the hunters could kill the whales by spearing them behind the flipper[45]. Similarly, single kayak-equipped eighteenth-century Unangan (Aleut) hunters of the Bering region used barbed non-toggling harpoons coated with aconite poison to immobilize the whales by spearing them near the flipper[46]. After a few days the whale could no longer remain upright and would drown and be towed to shore. While it is unlikely that aconite poison was part of the Saqqaq hunting strategy, a similar effect might have been achieved from harpoons infested with rotten meat or blubber, as even small flesh wounds can cause inflammation and, within days, immobilization of the flipper or death of such large whales[47]. Hence, using the Paamiut or the Aleut method, Saqqaq hunters armed only with penetrating lances and small harpoons may have been able to kill large, slow-swimming bowheads without Thule-style technology and large umiak boat crews.

The presented evidence of Saqqaq whale exploitation requires re-evaluating maritime history. Western history has always considered European whaling as the originator and pinnacle of marine exploitation, beginning with Basque whaling in the Bay of Biscay in the 1400s AD (refs 48,49). However, 1,500 years earlier, Inuit people of the Bering Strait region had developed technology sufficient for large whale hunting. The utilization of whale products thousands of years before the technology and communal organization of Thule whaling pushes back the first evidence of whale product usage in the Arctic and can be seen as a logical development of the powers of indigenous observation and ingenuity in the efficient use of a plentiful northern marine energy resource. We should not be surprised if Ocean Bay Kodiak Islanders, Early Jomon of Japan, and others around the Greater North Pacific Rim also found ways to use whale products long before purposeful whaling became a routine indigenous economic and social enterprise or a European Arctic industry.

Taken together, the high level of agreement between this DNA-based approach and previous morphological analyses along with the identification of previously unidentified species demonstrates that DNA deposited simultaneously with the fossils represents an equally important and complementary fraction of the faunal assemblage. We found that the subsistence contribution of some previously identified species such as caribou (Qajaa and Qeqertasussuk) and narwhal (Qajaa) had probably been underestimated by faunal analyses. Furthermore, the genomic

approach allowed us to identify several species for the first time, including the bowhead whale (Qajaa and Qeqertasussuk), walrus (Qajaa) and hooded seal (Qajaa). These findings expand our current knowledge of the Paleo-Inuit and illustrates that the Saqqaq people had a wider diet-breadth than was previously thought and were able to exploit most of the mammals available to them.

## Methods

**Sample collection.** The samples at Sandnes (64.242822, −50.177522, Lat/Long, WGS84) were collected in July 2013, using vertical drilling with a modified Stihl BT45 petrol drilling machine equipped with a diamond edged cylindrical drill bit 8 cm Ø and 10 cm long. At Fladstrand (74.0986500, −21.1872500, Lat/Long, WGS84) samples were collected from different profiles reported in Gotfredsen et al.[11], Sørensen et al.[12] and Jensen et al.[18]. Samples FA1-3 were collected from profile 100/203 as illustrated in the profile drawing in Jensen et al.[18], p. 18, and samples FB1-3 were collected from profile 104/209. Samples FC and FD were collected from 100/201 and 104/207, respectively (Supplementary Table 9). At Qajaa (69.127602, −50.702076, Lat/Long, WGS84) sampling was carried out from the 2nd to the 6th of August 2009. Samples from profile A were collected from the original profile from Møhl et al.[8] and Meldgaard[13]. For profile B, samples were collected from a test pit established behind excavation area 'D' in Meldgaard[13]. At Qeqertasussuk (68.5927333, −51.0719972. Lat/Long, WGS84) sampling were carried out the 9th of August 2009 at an area right behind 'area B' from Meldgaard[5].

All sediment samples were collected for the purpose of ancient DNA analyses and thus precautions were taken to prevent contamination from modern DNA during sampling. Handling of the samples was carried out wearing gloves and sediment containers were sealed immediately after sampling. For samples collected from already established profiles at Fladstrand and Qajaa, the exterior 20–30 cm of the profile was carefully removed before samples were collected by pushing falcon tubes into the profile. Sampling by vertical drilling at Sandnes and Qeqertasussuk was carried out using a generator-powered drilling machine, sealing off samples in Ziploc bags or PVC tubes immediately after drilling. All the samples analysed were collected from protected archaeological sites as part of approved archaeological excavations.

**Dating.** For accelerator mass spectrometry (AMS) datings, four samples from plant macro remains and a single bone fragment were collected from four sediment samples at Sandnes: V51-3; V51-5; V51-7; and V51-10 (Supplementary Table 9). AMS $^{14}$C analyses were carried out at Aarhus AMS Dating Centre on the five samples (Supplementary Table 10). $^{14}$C ages are reported in conventional radiocarbon years BP (before present = 1950 AD). All dates were calibrated with the calibration curve IntCal13 described in ref. 50 using Oxcal v4.1 (ref. 51). The probability measures reported in Supplementary Table 10 for calibrated age ranges, represents 68.2% probability (1 sigma) and 95.4% probability (2 sigma), respectively. Calibrated AMS results date the midden layers at Sandnes to have been deposited in the thirteenth century, corresponding to unit FI-5 in McGovern et al.[9]

**Total shotgun libraries.** All pre-PCR amplification steps were carried out in dedicated ancient DNA clean laboratories at the Centre for GeoGenetics, University of Copenhagen applying strict aDNA practices[52,53]. For each batch of extractions a minimum of two extraction blanks were included to which no sediment was added (Supplementary Table 11). Extractions for shotgun sequencing were carried out using a phenol–chloroform-based extraction protocol optimized for ancient sediments[54]. To minimize risk of contamination, the outer layers of each sediment sample were removed with disposable sterile scalpels and only the centres of each sample were subsampled for DNA extractions. For DNA extractions, ∼1–3 g of sediment was dissolved in 3 ml digestion buffer (18 mM EDTA, 100 µg ml$^{−1}$ proteinase K, 7% N-lauryl-sarcosyl, 50 mM dithiothreitol and 3% mercaptoethanol), homogenized for 2 × 20 s on a FastPrep-24 (MP Biomedicals) and incubated overnight at 37 °C. Following incubation, samples were spun down and the supernatant transferred to a clean 15 ml tube. Next, samples were subjected to an inhibitor removal step using buffers C2 and C3 from PowerMax Soil DNA Isolation Kit (MO BIO Laboratories, Inc., Carlsbad, CA, USA) followed by DNA isolation using phenol and chloroform in a ratio of 1:2. Finally, extracts were concentrated and purified with a 15 ml 30 kDa Amicon Ultra-4 Centrifugal Filter (Millipore) using two wash steps with 750 µl buffer C6 from PowerMax Soil DNA Isolation Kit. Extracts were stored at −20 °C in 50–100 µl C6 buffer. Libraries were built using the NEBNext DNA Library Prep Master Mix for 454 (E6070) as in Orlando et al.[55], with the following modifications: 1–20 µl extract was used for the end-repair step depending on the DNA concentration and inhibition level. End-repair reactions were incubated for 20 min at 12 °C and 15 min at 37 °C. At the adapter ligation step, reactions were incubated at 20 °C for 15min. The final fill-in reaction was performed at 60 °C for 20 min followed by 80 °C for another 20 min to inactivate the enzyme. Each

reaction step was followed by a purification step with MinElute columns (Qiagen) using 400 µl Buffer PB for each 25 µl reaction.

Index amplifications were carried out in a single PCR step with Illumina InPE 1.0 universal forward primer (5′-AATGATACGGCGACCACCGAGATCTACA CTCTTTCCCTACACGACGCTCTTCCGATCT-3′) in conjunction with a custom made indexed reverse primer (5′-CAAGCAGAAGACGGCATACGAGATNNN NNNGTGACTGGAGTTC-3′, where the N stretch corresponds to a six nucleotide index tag)[56]. PCR cycling conditions consisted of an initial denaturation for 4 min at 94 °C followed by 14–16 cycles of 30 s at 94 °C, 30 s at 60 °C and 20 s at 68 °C, and a final elongation step for 7 min at 68 °C. The majority of amplified libraries were purified with the AMpure XP system (Agencourt), while a part of the libraries were purified with MinElute columns (Qiagen). Following purification, all libraries were quantified on a Bioanalyzer 2,100 (Agilent) using the High-Sensitivity DNA Assay kit. To evaluate potential contaminations during the library build process, blank libraries with no DNA input and index PCR blanks were included in the workflow. Finally, libraries were pooled in equimolar concentrations and sequenced on the Illumina HiSeq platform in 100 bp paired-end mode.

**Helminth shotgun libraries.** DNA extraction and library preparation was performed at the dedicated ancient DNA laboratory at Centre for GeoGenetics. DNA extraction was performed using the PowerLyzer PowerSoil DNA isolation kit (MO BIO Laboratories, Carlsbad, California) with the following exceptions: bead beating was performed with Lysing Matrix I beads (MP Biomedicals, Santa Ana, California) for 90 sec at 6.5x on the FastPrep-24 (MP Biomedicals) and DNA was eluted in 60 µl elution buffer. Blunt-end DNA libraries were prepared as described above, although intermittent reaction clean-up was performed using the MinElute PCR purification kit (Qiagen), with an improved binding buffer that has proved highly efficient in recovering very short DNA fragments[57]. Index amplifications were performed using a nested PCR approach using 5 U Taq Gold (Life Technologies), 1× buffer Gold, 2 mM MgCl$_2$, 0.4 µg µl$^{-1}$ BSA, 0.25 mM dNTPs and 0.075% dimethylsulphoxide for each reaction with the following PCR conditions: an initial denaturation for 8 min at 94 °C followed by 12 cycles of 30 s at 94 °C, 30 s at 60 °C and 40 s at 72 °C, and a final elongation step for 7 min at 72 °C. The first reaction round consisted of 50 µl reactions using the entire DNA library as template for 12 cycles using Illumina in PE1.0 and custom-made indexed reverse primers, as described above. Second-round amplification (25 µl) was performed using 5 µl of first round PCR product as template and Illumina P5 (5′-AATGA TACGGCGACCACCGA-3′) and P7 primers (5′-CAAGCAGAAGACGGCATA CGA-3′) for 10–12 cycles. PCR cycling and post-PCR handling was performed in DNA laboratories physically separated from the aDNA laboratories. Second-round PCRs were visually investigated on a 2% agarose gel, then purified using the MinElute kit following the manufacturer's instructions, and quantified on a Qubit 2.0 using the dsDNA HS kit (Thermo Fischer) and finally on a Bioanalyzer (Agilent Technologies) using the Agilent High Sensitivity DNA kit. Purified libraries were pooled at concentrations of 5–20 nM before sequencing using 100 bp single read chemistry on a HiSeq 2000/2500 platform at The Danish National High-Throughput DNA Sequencing Centre.

**Data pre-processing.** DNA sequencing reads were subjected to several steps of pre-processing. First, raw read files were demultiplexed with Novobarcode (Beta-0.8). Next, reads were quality trimmed and adapter sequences removed using AdapterRemoval (v. 1.5.4)[58], where read pairs with overlapping 3′ sequences longer than 11 nucleotides were collapsed into single reads. Sequences shorter than 25 bases were discarded, base quality threshold was set to 25 and stretches of Ns were trimmed at both ends of the reads. To remove low-complexity reads, sequences were filtered using sga preprocess (https://github.com/jts/sga), discarding reads with a dust score higher than 1. Finally, PCR duplicates were removed using a custom-made python script that discards reads if they are exact copies of already existing reads (that is, same sequence and same length).

**Taxonomic assignment.** A set of python scripts was developed to assign reads to taxonomic nodes based on a lowest common ancestor approach (https://github.com/frederikseersholm/getLCA). Briefly, filtered, trimmed and collapsed reads were mapped against the NCBI databases of complete reference genomes of mitochondria in *Metazoa* and plastids in plants (http://www.ncbi.nlm.nih.gov/genome/organelle/) using Bowtie2 (2.2.4)[59]. To identify each entry in the databases, taxids were added to the fasta header for the databases using a python script parsing the NCBI file gi_taxid_nucl.dmp (ftp://ftp.ncbi.nih.gov/pub/taxonomy/gi_taxid_nucl.dmp.gz). To retain only high-confidence hits, alignments with an edit distance above 5% of the read length were discarded. Next, each read was assigned to the taxonomic node corresponding to the lowest common ancestor of the best hit(s) in the database, based on the NCBI taxonomy files: names.dmp and nodes.dmp. (ftp.ncbi.nih.gov/pub/taxonomy/taxdump.tar.gz). That is, for reads with several equally good alignments, taxonomic assignment was achieved by assigning the read to the lowest common ancestor of these hits, while reads with a single best alignment could be assigned directly to the taxonomic node of the best hit.

To visualize data in Figs 2 and 3 and Supplementary Table 8, higher-order taxa represented by a single species in the data set were collapsed to the species as

follows: *Bos* and *Bovinae* were collapsed to *Bos*; *Canis*, *C. lupus* and *C. lupus familiaris* were collapsed to *C. lupus*; *Pusa* and *P. hispida* were collapsed to *P. hispida*; *Balaenidae* and *B. mysticetus* were collapsed to *B. mysticetus*; and *Cervidae*, *Odocoilinae* and *R. tarandus* were collapsed to *R. tarandus*.

**Biomass estimates.** Biomass estimates in Fig. 3 are based on estimates from Table 9.5 in ref. 5 (Supplementary Table 7). For a thorough description, we refer to the original source. Minimum number of individuals (MNI) counts for each species were estimated from the most frequently occurring indicator bone; mammals: the mandible (except for fox in layer 3; humerus); birds: the humerus. Bone counts were based on a sample of 407 kg of bone fragments excavated from areas B and C at Qeqertasussuk. This sample represents ∼8.7% of the total, both unexcavated and excavated, and the MNIs have accordingly been multiplied by a factor 11.5. The age category for the indicator bone (mandible) from ringed seals and harp seals have been calculated on tooth section analyses (see Tables 9.6 and 9.8 in ref. 5). *sed*aDNA read counts presented in Fig. 3 represents DNA read counts from Qeqertasussuk for harp seal (*P. groenlandicus*), ringed seal (*Pusa and P. hispida*) fox (*Vulpes*), as well as all bird species identified (*Laridae*, *Larus* and *Anatidae*).

**Data availability.** DNA sequencing data for vertebrate and plant DNA from this study have been deposited at EBI under study accession PRJEB13329. All other data can be requested from the authors.

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

## Acknowledgements

We thank Bjarne Grønnow for his advice and expertise concerning Saqqaq culture archaeology. Furthermore we thank Anders Anker Bjørk for providing the map of Greenland presented in Fig. 1. This work was supported by the Danish Advanced Technology Foundation, the University of Copenhagen 2016 Initiative 'The Genomic History of Denmark' and the Villum Foundation through the pilot project 'Alle Tiders Mennesker'. Special thanks to the Greenland National Museum & Archive in Nuuk, Christian Koch Madsen and Ann Eileen Lennert.

## Author contributions

M.W.P. initiated the study and conceived the idea; collected samples; designed, performed and interpreted experiments; supervised molecular analyses and co-wrote the manuscript. F.V.S. prepared NGS libraries for metagenomic analyses; designed and implemented the computational pipeline to analyse the data; designed and interpreted experiments and drafted the manuscript and figures. M.J.S. developed the concept, designed, performed and interpreted analyses of parasitic helminths and co-wrote the manuscript. S.S.T.M. and H.S. prepared libraries for metagenomic analyses. A.R. analysed macrofossils for C-14 dating from Sandnes and contributed to the manuscript. K.H.K., M.R. and E.W. contributed to the manuscript and collected samples. W.F. provided interpretation of the archaeological context and contributed to the discussion. M.M. conceived the idea, collected samples, interpreted data and provided interpretation of the archaeological context. C.M.O.K. oversaw the study of parasitic helminths, designed, analysed and interpreted these experiments and co-wrote the manuscript. A.J.H. lead and oversaw the study, designed, analysed and interpreted experiments and co-wrote the manuscript.

## Additional information

**Competing financial interests:** The authors declare no competing financial interests.

**Publisher's note**: 

