## [Peer Review File · Nature Communications]

Reviewers' comments:

Reviewer #1 (Remarks to the Author):

Review of Seersholm et al "DNA evidence of bowhead whale exploitation by Greenlandic Paleo-Inuit 4000 years ago".

This is an interesting paper which using sedimentary ancient DNA to identify bowhead whale exploitation by Paleo-Inuit.

I found this paper enjoyable to read and of high general interest.

I have only a few relatively minor comments for the authors.

1. The read counts for the vertebrate species are, not surprisingly low, apart from the Bow head whale and seal signatures. The authors cite relevant papers regarding low level vertebrate contaminants in standard reagents, but fail to list extraction blank details in their supplementary tables. Most are not relevant, however dog DNA, as well as other standard domesticates are readily seen - so I wonder if a strong caveat on the 2, 3 (highest 29) Canis reads should be mentioned?
2. Definitive ID of wolf and non-domesticated dog (from 7-41 reads) seems premature. Can the authors cite specifically how this was made, it may be more prudent here to use the genus rather than specialization as it is exceptionally difficult to discern the two, even in some cases from whole genome data.
3. Cow, sheep and goat are also common contaminants-perhaps again worthy of a note in the ms on their alternative possible origin?
4. On page 5 lines 187-190, the statement that "the high level of DNA damageat the two sites.", seems like a bit of a stretch. How can one be certain these signatures are true for all vertebrate species. While the whale and seal DNA signatures are clear cut do the authors feel this is sufficient to suggest all signatures are indeed real and from the time of occupation, i.e. no leaching?
5. Page 5 line 195. This is not a unique assessment, there are other papers where this has been done - suggest rewording and proper citations to give credit where due.
6. Authors propose, interestingly, to infer correlations of biomass and possible bone counts with DNA read numbers. This seems very precarious as preservation is likely to depend severely on tissue type and microenvironments in the sediment which are very difficult to model. Can the authors place something in the SOM which chronicles exactly how biomass was calculated - the only reference is to a not easily accessible Danish publication- and it's link with DNA read count? Thanks.
7. While interesting I find the plant DNA data not particularly relevant to the current paper. It is unclear what it adds - suggest possibly placing it entirely in the SOM?

Reviewer #2 (Remarks to the Author):

The authors present a study on ancient DNA sequencing analysis of four midden deposits of the Saqqaq culture in Greenland. They compared the results to the fossils from the sites and thereby revealed that also animals not represented in the fossil records, in particular walrus and whale species, played a major role in the subsistence of the Inuit. Moreover, they report the first evidence of bowhead whale exploitation in the Saqqaq culture.

This is a very interesting and well done study. It deals with a very important topic, whether or how far a fossil record represents the variety of animals used for the subsistence of a certain community or population. It seems to be quite obvious that not all hunted animals were carried back to the village as

a whole, in particular large animals such as bears and elks but also walrus and whales. The absence of such animals in the fossil records leaves open the question whether these species were hunted or exploited at all. Modern sequencing technologies now open a new access to these question by detecting ancient DNA residues that allow to extend our knowledge on the utilisation of various animal sources.

The manuscript should be of great interest for the reader of this journal and the scientific community. There a just a few issues that should be addressed by the authors before publication:

General comments:

1. It would be interesting to know if there are any data available on the human populations that lived at the studied sites. In particular, if there are any stable isotope data that would give insights into the overall nutrition of the Inuit. I am aware that there are some stable isotope data from the permafrost preserved hair of the Saqqaq culture which was published by the same group (Rasmussen et al., Nature 2010). Are there any other data available? Are the differences in faunal subsistence between the cultures also reflected in the human remains? The authors should comment on it.
2. I like the idea to correlate the helminths DNA with the specific hosts. Is there anything known on the prevalence of these parasites in the various animals? Are they e.g. more common in domesticated animals (dogs?) compared to wild animals? Could this influence the analysis? Please comment.
3. Did the authors identify any helminth species that are considered as human pathogens? Could there be deduced any risks for parasite infections due to the consumption or close proximity to these animals?
4. It is not clear whether the plant DNA reflect the overall vegetation of the sites or mainly mirrors a selection of plants that were part of the nutrition of the population. This should be clarified.

Specific comments:

1. Page 3, lines 96-97: "enriched for helminth eggs" is a poor choice of word. It could be confused with targeted in-solution enrichment. Furthermore, the method of naming "helminth shotgun libraries" is strange as both are shotgun libraries. The former is sedaDNA from soil and the latter is sedaDNA of helminth eggs
2. page 9, line 324: "advocates" is a strange word choice
3. page 12, lines 477-478: "exact match duplicates" do you mean PCR duplicates?
4. Page 12, line 481: assignment should be used instead of "assignation". Please correct throughout the text and supplementary information.
5. Page 12, line 481: Python scripts should be made publically available on github.com with a link provided in the text. Furthermore, if applicable, their dependencies should be referenced.

Supplementary Information:

1. Supplementary Discussion 2: "Primates and Phasianidae was identified as contamination and discarded"... If blanks were sequenced it should be easy to quantify the amount of lab based human contamination in each library (see supplementary of Cassidy et al 2015). It may be worth including a mapdamage analysis on the human reads if there are authentic human remains co-extracted too?
2. Supplementary Discussion 2: "Such false positives are presumed to be an effect of DNA damage or sequencing errors causing a read to resemble a close relative to the true species." One could remove the majority of deaminations by trimming the 5 and 3 ends by 2 bp and re-analyzing the reads.
3. Supplementary Figure 1: "Vegetation cover" I don't think sequencing data from middens are proper proxies for vegetation cover in the landscape. I would use a more tentative title like abundance
4. Supplementary Figure 2: a) Coverage plot: The green bar appears superfluous given the mean

coverage is indicated on the graph; b) The meaning of arrows in the deamination graphs is unclear to me. The 5' arrow is pointing away from its prime end, while the 3' is pointing toward. The numbers on the X-axis are sufficient to indicate the directionality of the DNA 5' and 3' end.

5. Supplementary Table 7. Column Protocol: They are both shotgun library protocols. The difference is with sampling type; helminth egg or sediment

Reviewer #3 (Remarks to the Author):

In their article entitled 'DNA evidence of bowhead whale exploitation by Greenland Paleo-Inuit 400 years ago', Seersholm et al. present new evidence for the early exploitation of bowhead whale within Paleo-Inuit sites in Greenland. Through the analysis of sedaDNA from midden deposits at four well-characterized archaeological sites, spanning multiple cultural occupations, they substantiate the range of taxa detected through previous zooarchaeological analyses, and detect previously un(der)represented species (bowhead whale, walrus, and caribou) as well as characterize plant communities at the site. The DNA analysis is completed to a high standard, and the authors follow published standards for the authentication of their data.

The authors assert that the detection of concentrated bowhead whale DNA within two Paleo-Inuit sites represents the most notable finding of the paper. Indeed the high proportion of bowhead whale DNA is intriguing, considering the lack of associated cetacean bone material, and does represent the first evidence of whale exploitation by Paleo-Inuit in Greenland. However, there is some question as to whether the exploitation of whales would be considered unexpected in terms of our understanding of the human behavior of coastal peoples. Whale bone is abundant in many Northwest coast and Holarctic archaeological sites, and there is a long history of exploitation of cetacean resources worldwide dating back at least 10,000 years. More than 70 years ago Graham Clark (1947, *Antiquity* 21, 84-104) described the Neolithic use of whale in the economy of Northern Europe, noting zooarchaeological evidence for the widespread use of stranded whales, but also iconography supporting very early whale hunting (a publication which should be cited in Seersholm et al's paper). It is also well established that cetaceans in particular are poorly represented in archaeological sites, due to the difficulty in transporting large carcasses from shore to a habitation site (as discussed in Smith and Kinahan 1984, *The invisible whale*, *World Archaeology*, 16, 89-97 and Szabo 2008, *Monstrous Fishes and the mead-dark sea*, and touched upon by the authors). Likewise, ethnographic evidence (predominantly from the Northwest Coast of North America) suggests a focus on blubber exploitation (with blubber from drift whales being highly prized), and to a lesser extent on meat, and with almost no evidence for the exploitation or use of whale bones (e.g., Monks 2001, *Int. J. Osteoarchaeol.* 11: 136-149) Therefore, the presence of whale sedaDNA but the absence of whale bone within the middens is interesting, but perhaps unsurprising in terms of our knowledge of coastal resource exploitation.

Moreover, the scale of exploitation is not conclusively demonstrated by the DNA evidence. The authors note that for harp and fur seal there are good correlations between bone biomass and the proportion of DNA reads (where sedaDNA could be argued to be principally derived from the degradation of bone and relatively smaller quantities of discarded meat, fat, skin, etc.). However, extrapolating this relationship to large cetaceans is problematic, since the sedaDNA from whale is obviously derived almost exclusively from non-bone products (which may more readily leach host DNA into the soil). Ethnographic evidence suggests that whale blubber (and skin) was often boiled to render the oil, which was skimmed off. Discarding the waste water in the midden might account for a high concentration of whale sedaDNA, and perhaps even a high degree of DNA leaching through the midden. Interestingly, the authors found that the calculated biomass of domestic animals at Sandnes was three times higher than that expected from osteological evidence, noting that feces and urine can

concentrate DNA in the archaeological record. This result reinforces the difficulties of resolving 'actual' biomass used, based on sedaDNA results. To strengthen the manuscript, the authors could be far more critical of their argument concerning expected relationships between bone biomass and sedaDNA concentrations.

A high proportion of whale sedaDNA is also not strong evidence for whale hunting as opposed to whale scavenging. Archaeological investigations at sites which span time periods involving the shift from scavenging to active hunting show limited differences in terms of the quantity of whale bone at a site, even though active whale hunting is thought to result in a greater quantity of whale biomass (e.g. Monks 2001, *Arctic Anthropology*, 38, 60-81). Although the authors note that the observed sedaDNA results may be a result of scavenging draft carcasses, there is a focus on whale hunting in the manuscript, even though the former explanation is the most parsimonious. More discussion on the widespread exploitation of stranded cetacean carcasses (observed in virtually all coastal economies where cetaceans strand and which continues even after the development of active hunting) would provide a more balanced interpretation of the results.

It is perplexing that although complete mtDNA genomes of bowhead whale were constructed to a relatively high depth of coverage, the authors did not attempt to distinguish the minimum number of individuals that might be represented by the DNA sequences. Based on the data currently provided in the paper, for both middens it appears that the DNA sequences derive from only a single whale, respectively. A finding of several, distinct haplotypes present within the sedaDNA (e.g., through mtDNA genome capture), would provide stronger evidence for the continued and extensive exploitation of whale products. Following from that point, although I wholeheartedly agree with the authors that historians and archaeologists frequently underestimate the maritime technologies and capabilities of native cultures, the evidence provided in this paper can only provide the first concrete evidence for the early exploitation of whale products by the Paleo-Inuit, but is not particularly convincing of active hunting. The latter conclusion would need to be supported by far more evidence for the exploitation of multiple whale individuals that surpassed expectations for bowhead whale strandings in the area.

Although the analysis of sedaDNA to document the use of animals within archaeological sites or palaeosols is now well established (particularly within arctic or Holarctic contexts), the use of shotgun sequencing to detect host-specific parasites is a novel approach for documenting taxonomic diversity within archaeological sites, and an interesting complement to zooarchaeological and sedaDNA approaches.

In sum, this is an interesting paper, with robust and novel sedaDNA data important for understanding animal exploitation and economies in Greenland. At the moment, too much emphasis is placed on the 'notable' detection of whale sedaDNA, which undermines some of important data on other identified taxa (including plants), and the novel use of parasite sedaDNA to understand species distributions. This paper makes an important contribution to archaeological understandings of subsistence economies within Greenland, but would perhaps be more appropriate for a specialized journal in zooarchaeology or Holarctic archaeology.

General remarks to reviewers

We would like to thank the reviewers for the very high quality of their comments. They touch upon almost all of the issues we have been discussing vividly during the writing of this manuscript. We have addressed all of the comments and believe that the revised version of the manuscript represent a substantial improvement from the previous form.

Reviewer #1 (Remarks to the Author):

Review of Seersholm et al "DNA evidence of bowhead whale exploitation by Greenlandic Paleo-Inuit 4000 years ago".

This is an interesting paper which using sedimentary ancient DNA to identify bowhead whale exploitation by Paleo-Inuit.

I found this paper enjoyable to read and of high general interest.

I have only a few relatively minor comments for the authors.

1. The read counts for the vertebrate species are, not surprisingly low, apart from the Bow head whale and seal signatures. The authors cite relevant papers regarding low level vertebrate contaminants in standard reagents, but fail to list extraction blank details in their supplementary tables. Most are not relevant, however dog DNA, as well as other standard domesticates are readily seen - so I wonder if a strong caveat on the 2, 3 (highest 29) Canis reads should be mentioned?

We completely agree with the reviewer on this point and have included control samples in the SI (Supplementary Table 6). We do not identify any DNA from sheep and goat. However, as depicted in the table, we do identify three *Canis lupus* reads in the extraction blank from the helminth library preparation of samples from Sandnes (S-P-EB). While we don't identify *Canis lupus* reads in blanks from the sediment library preparation batch, this could indicate that *Canis lupus* reads at Sandnes represent a contamination. On the other hand, we do not identify canid DNA in any of the peat samples at Sandnes (S1-P, S1, S2 and S3).

To be conservative, we have marked the *Canis* reads at Sandnes as potential contamination and removed them from Figure 2.

2. Definitive ID of wolf and non-domesticated dog (from 7-41 reads) seems premature. Can the authors cite specifically how this was made, it may be more prudent here to use the genus rather than specialization as it is exceptionally difficult to discern the two, even in some cases from whole genome data.

The identification of *Canis* reads is based on the lowest common ancestor approach used to characterize all of the reads in this study. The mitochondrial reference

database contains 7 mitochondrial genomes from the *Canis* genus, of which 5 are subspecies of *Canis lupus*: *Canis lupus lupus* (NC_008092.1), *Canis lupus lupus* (NC_009686.1), *Canis lupus chanco* (NC_010340.2), *Canis lupus familiaris* (NC_002008.4) and *Canis lupus laniger* (NC_011218.2). Reads with the same edit distance (i.e. number of mismatches) between two different subspecies of *Canis lupus*, say, *Canis lupus lupus* and *Canis lupus familiaris*, will be assigned to species level (*Canis lupus*). Reads aligning to a the reference sequence of a single subspecies with a lower number of mismatches than any of the other subspecies (or any other mitogenome in the database) will be assigned directly to that subspecies.

At Fladstrand for example, 93 reads were assigned within the *Canis* genus. Of these, 29 were assigned to genus level (*Canis*), 41 to species level (*Canis lupus*) and 21 to the dog subspecies (*Canis lupus familiaris*). In addition one read respectively was assigned to *Canis lupus lupus* and *Canis lupus chanco*.

3. Cow, sheep and goat are also common contaminants-perhaps again worthy of a note in the ms on their alternative possible origin?

Corrected. We have added the following comment on this in Supplementary Discussion 2:

“While sheep and goat have been identified previously as common contaminants, we do not identify these species in either peat or control samples.” SI p.12.

4. On page 5 lines 187-190, the statement that “the high level of DNA damageat the two sites.”, seems like a bit of a stretch. How can one be certain these signatures are true for all vertebrate species. While the whale and seal DNA signatures are clear cut do the authors feel this is sufficient to suggest all signatures are indeed real and from the time of occupation, i.e. no leaching?

To avoid overinterpreting the DNA damage, these sentences have been reworded to focus only on harp seal, caribou and bowhead whale, from the two libraries QA6 and QT3:

“Furthermore, the high level of DNA damage for mammal species at Qajaa (harp seal and caribou) and Qeqertasussuk (harp seal, caribou and bowhead whale), confirms that these signals represent authentic DNA deposited at the time of occupation at the two sites.” p.4 l.206-209

However, we strongly believe that the remaining mammal species do represent endogenous ancient DNA as well. The absence of mammal DNA in the peat layer at Qajaa confirms that leaching from the above Dorset layer is negligible and the absence of contamination from relevant species (seals, whales, walrus and caribou) demonstrates the ancient nature of the DNA analysed. In addition, a previous study at a similar site in Greenland did not find evidence of leaching¹.

[1] Hebsgaard, M. B. *et al.* ‘The Farm Beneath the Sand’- an archaeological case study on ancient ‘dirt’ DNA. *Antiquity* **83**, 430–444 (2009).

5. Page 5 line 195. This is not a unique assessment, there are other papers where this has been done - suggest rewording and proper citations to give credit where due.

The sentence has been rephrased and now reads:

“Apart from the presumed Dorset layer at Qajaa, bone fragments recovered from all midden layers in the present study have previously been examined using morphological techniques, thus allowing for a comparative assessment of the aDNA performance similar to comparisons reported in previous studies²⁻⁴”
p.4 l.211-215.

6. Authors propose, interestingly, to infer correlations of biomass and possible bone counts with DNA read numbers. This seems very precarious as preservation is likely to depend severely on tissue type and microenvironments in the sediment which are very difficult to model. Can the authors place something in the SOM which chronicles exactly how biomass was calculated - the only reference is to a not easily accessible Danish publication- and it's link with DNA read count? Thanks.

Corrected. We have added a description of the MNI estimation from Meldgaard 2004 in ‘Supplementary Methods 3 – Biomass estimates from Meldgaard 2004’:

“Supplementary Methods 3 – Biomass estimates from Meldgaard 2004

*Biomass estimates in Figure 3 are based on estimates from table 9.5 in Meldgaard 2004⁵ (see below). For a thorough description, we refer to the original source. MNI counts for each species were estimated from the most frequently occurring indicator bone; mammals: the mandible, (except for fox in layer 3; humerus), birds: the humerus. Bone counts were based on a sample of 407 kg of bone fragments excavated from Area B, and C at Qeqertasussuk. This sample represents approximately 8.7% of the total, both unexcavated and excavated, and the MNI's have accordingly been multiplied by a factor 11.5. The age category for the indicator bone (mandible) from ringed seals and harp seals have been calculated on tooth section analyses (see table 9.6 and 9.8 in Meldgaard 2004⁵). sedaDNA read counts presented in Figure 3, represents DNA read counts from Qeqertasussuk for harp seal (*Pagophilus groenlandicus*), ringed seal (*Pusa* and *Pusa hispida*), fox (*Vulpes*) as well as all bird species identified (*Laridae*, *Larus* and *Anatidae*).”* SI p.15

	Age category	Live weight (kg)	MNI	Biomass (kg)	Abundance (%)
Arctic fox		3.00	903	2709	0.9
Ringed seal	0-3 yrs	25.00	1764	44100	15.3
	3 - yrs	35.00	352	12320	4.3
Harp seal	0-1 yrs	30.00	920	27600	9.5
	1-4 yrs	75.00	922	69150	23.9

	5- yrs	130.00	884	114920	39.7
Fulmar	0.75	6630	4972.5	1.7	
Ptarmigan	0.50	1168	584	0.2	
Gulls	1.50	1914	2871	1	
Little auk	0.15	1634	245.1	0.1	
Brunnich's guillemot	1.10	8769	9645.9	3.3	
Total		25860	289117.5	100	

Supplementary Table 1 | MNI estimates from Qeqertasussuk. The table represents the estimates presented in column 3 (All faunal components) from table 9.5 in Meldgaard 2004.

Additionally we have addressed some of the potential biases when inferring biomass from DNA sequencing data in “Supplementary Discussion 3 – Inferring biomass from DNA profiles” SI p. 12-13.

7. While interesting I find the plant DNA data not particularly relevant to the current paper. It is unclear what it adds - suggest possibly placing it entirely in the SOM?

We agree that the findings of the plant section do not provide any valuable findings in terms of past vegetation changes or subsistence plants. Merely, this section serves as a proof of concept. As, reviewer#3 suggests to put more emphasis on other reported taxa, including plants we have left this paragraph in the main text for now. However, if requested by reviewers we are more than willing to move the plant paragraph to the supplementary material.

Reviewer #2 (Remarks to the Author):

The authors present a study on ancient DNA sequencing analysis of four midden deposits of the Saqqaq culture in Greenland. They compared the results to the fossils from the sites and thereby revealed that also animals not represented in the fossil records, in particular walrus and whale species, played a major role in the subsistence of the Inuit. Moreover, they report the first evidence of bowhead whale exploitation in the Saqqaq culture.

This is a very interesting and well done study. It deals with a very important topic, whether or how far a fossil record represents the variety of animals used for the subsistence of a certain community or population. It seems to be quite obvious that not all hunted animals were carried back to the village as a whole, in particular large animals such as bears and elks but also walrus and whales. The absence of such animals in the fossil records leaves open the question whether these species were hunted or exploited at all. Modern sequencing technologies now open a new access to

these question by detecting ancient DNA residues that allow to extend our knowledge on the utilisation of various animal sources.

The manuscript should be of great interest for the reader of this journal and the scientific community. There a just a few issues that should be addressed by the authors before publication:

General comments:

1. It would be interesting to know if there are any data available on the human populations that lived at the studied sites. In particular, if there are any stable isotope data that would give insights into the overall nutrition of the Inuit. I am aware that there are some stable isotope data from the permafrost preserved hair of the Saqqaq culture which was published by the same group (Rasmussen et al., Nature 2010). Are there any other data available? Are the differences in faunal subsistence between the cultures also reflected in the human remains? The authors should comment on it.

Apart from in Rasmussen et al, the bones from Qeqertasussuk have been described in Koch et al², although no stable isotope analysis were carried out. However several recent studies have compared diet from the Norse and the Inuit using stable isotope analyses^{3,4}. These studies suggest an extreme reliance on a marine diet for the Inuit, while for the Norse, subsistence was based on both domestic and marine animals in comparable quantities. Furthermore, it is suggested that the Norse diet shifted to consist predominantly on marine food towards the end in the 14th century, as the temperatures fell during the Little Ice Age.

[2] Koch, A., Frølich, B., Lynnerup, N., and Hart Hansen, J. P. 1996. The bones from Qeqertasussuk: The earliest human remains from greenland. In: Grønnow, B. ed. The Paleo-Eskimo cultures of Greenland: New perspectives in Greenlandic archaeology. Danish Polar Center Publications, No. 1. Copenhagen: The Danish Polar Center. 35-37.

[3] Nelson, D. E. Heinemeier, J., Lynnerup N, Sveinbjørnsdóttir, A. E. Arneborg J. 2012 An isotopic analysis of the diet of the Greenland Norse. Journal of the North Atlantic 3: 93 118.

[4] Stable Isotopes and Oral Tori in Greenlandic Norse and Inuit: Baumann, Mathilde ; Lynnerup, Niels ; Richard Scott, G. International Journal of Osteoarchaeology, 05/2016

We have elaborated on this on lines 309-316 in the manuscript:

“The ordination analysis clearly differentiates between the diet based on both domestic and marine animals for the Norse and the extreme reliance on marine resources for the Inuit compared to (Supplementary Fig. 5). This difference in subsistence practices is also reflected in stable isotope data from human bone remains from Saqqaq³⁰, Dorset³¹, Thule³² and Norse^{32,33} cultural sites. Analysis of isotope composition in Inuit remains suggests a strong dependence on marine resources, while the Norse bone remains show evidence of subsistence based on both domestic and marine animals in comparable quantities.” p.6 l.309-316.

2. I like the idea to correlate the helminths DNA with the specific hosts. Is there anything known on the prevalence of these parasites in the various animals? Are they e.g. more common in domesticated animals (dogs?) compared to wild animals? Could this influence the analysis? Please comment.

We agree with the reviewer that it would be very interesting if we could use the abundance of different helminth species to infer the most likely host species. However, the helminth DNA can hardly be linked to exact prevalence as the deposition on the site may rely on several factors eg. direct deposition by the host (dogs, sheep) or indirect if faeces is removed to a particular site (dung pile) by man. The Helminth data is primarily qualitative indicative for a particular lifecycle and associated hosts.

3. Did the authors identify any helminth species that are considered as human pathogens? Could there be deduced any risks for parasite infections due to the consumption or close proximity to these animals?

Yes, two helminth species identified (*E. canadensis* and *T. canis*) have zoonotic properties causing respectively liver cysts and larvae migrans, the first may be associated with severe clinical manifestation the latter mostly passing with very mild or none symptoms. Both are transmitted to man from faeces of carnivores, thus close proximity to dogs may have constituted an impact on health. We have highlighted this in the legend for Supplementary Table 4 (SI p. 6).

4. It is not clear whether the plant DNA reflect the overall vegetation of the sites or mainly mirrors a selection of plants that were part of the nutrition of the population. This should be clarified.

Thanks for the comment. To clarify this we have included a brief comment in the section 'Plant DNA':

"These plant families are consistent with the current vegetation cover in Greenland dominated by grasses (Poaceae) and low-lying shrubs such as dwarf willows (Salicaceae) and crow berries (Ericaceae). This suggests that the plant DNA identified here represents the vegetation cover at the midden for each habitation period rather than plants of significant value to subsistence." p.4 l.179-183.

Specific comments:

1. Page 3, lines 96-97: "enriched for helminth eggs" is a poor choice of word. It could be confused with targeted in-solution enrichment. Furthermore, the method of naming "helminth shotgun libraries" is strange as both are shotgun libraries. The former is sedaDNA from soil and the latter is sedaDNA of helminth eggs

The sentence have been rephrased and now reads:

"From these we generated 31 shotgun libraries based on the total DNA extracted from midden sediment (hereafter referred to as sediment libraries) and 27 shotgun libraries based on helminth eggs isolated from the sediment by sieving (hereafter referred to as helminth libraries)." p.2-3 l.93-96

2. page 9, line 324: *"advocates" is a strange word choice.*

Changed to *suggests*.

3. page 12, lines 477-478: *"exact match duplicates" do you mean PCR duplicates?*

Yes, changed to *PCR duplicates*.

4. Page 12, line 481: *assignment should be used instead of "assignation". Please correct throughout the text and supplementary information.*

Corrected throughout the manuscript and the SI.

5. Page 12, line 481: *Python scripts should be made publically available on github.com with a link provided in the text. Furthermore, if applicable, their dependencies should be referenced.*

Corrected. Scripts used in this study have been made publically available under: <https://github.com/frederikseersholm/getLCA>

Supplementary Information:

1. *Supplementary Discussion 2: "Primates and Phasianidae was identified as contamination and discarded"... If blanks were sequenced it should be easy to quantify the amount of lab based human contamination in each library (see supplementary of Cassidy et al 2015). It may be worth including a mapdamage analysis on the human reads if there are authentic human remains co-extracted too?*

We agree with the reviewer that the human DNA would provide a very interesting angle to this paper. However, with the identification of human DNA in control samples and the low numbers of human DNA reads identified in the sediment samples we do not have sufficient power to proof the ancient nature of these reads.

The estimation of background human DNA from Cassidy et al. 2015 requires that blank and test samples have been prepared with identical protocols. However, the blanks of this study have all been amplified for 16 cycles during index PCR in order to detect low level background contamination, while the sediment samples have been amplified with either 14 or 16 cycles dependent on initial DNA concentration and inhibition level. Hence, the quantification of contamination level as presented in Cassidy et al. is unfortunately not suitable for the samples in this study.

The level of human DNA in anthropogenic sediment samples usually varies between 5-50 reads, which is too low for reliable quantification of DNA damage patterns. The

library with the highest level of human DNA is the resequenced library from the middle layer at Qajaa (QA6), where we identify 160 reads assigned to *Homo* or *Homo sapiens*. As depicted below, the human reads from this layer is very likely to be of ancient origin. A capture approach would be required to confirm this. Unfortunately, this is out of the scope of this manuscript.

Figure 1. *Homo sapiens* DNA damage pattern from library QA6 based on 160 reads assigned to *Homo* or *Homo sapiens*.

2. Supplementary Discussion 2: "Such false positives are presumed to be an effect of DNA damage or sequencing errors causing a read to resemble a close relative to the true species." One could remove the majority of deaminations by trimming the 5 and 3 ends by 2 bp and re-analyzing the reads.

Thanks for this suggestion – as outlined below this approach provided a strong support for our hypothesis. Two trim sizes were tested: 2bp and 5bp. Using these trimming settings we detect a reduction in reads assigned to the 4 expected false positives *Phoca largha*, *Pusa sibirica*, *Pusa caspica* and *Phoca fasciata* from 68 reads in total to 44 and 24 reads in total for 2bp and 5bp trimming, respectively. This suggests that a large fraction of these assignments can be explained by 5' C to T misincorporations. However, with these trimming settings we loose valuable data, as a fraction of the assigned reads – 7.1% and 27.2% of all vertebrate reads for 2bp and 5bp, respectively – drops below the size threshold and is excluded from the analysis.

Based on these results, we have decided to retain as much data as possible by not applying any trimming, while discussing the results from the trimming analysis in the SI:

*This hypothesis was tested by trimming all reads from each end with two trim sizes: 2bp and 5bp. Using these trimming settings we detect a reduction in reads assigned to the 4 expected false positives *Phoca largha*, *Pusa sibirica*, *Pusa caspica* and *Phoca fasciata* from 68 reads in total to 44 and 24 reads in total for 2bp and 5bp trimming from each end, respectively. This suggests that a large fraction of these assignments can be explained by 5' C to T misincorporations. However, with these trimming settings we loose valuable*

data, as a fraction of the assigned reads – 7.1% and 27.2% of all vertebrate reads for 2bp and 5bp, respectively – drops below the size threshold. Based on these results, we have decided to retain as much data as possible by not applying any trimming. SI p.12-13

3. Supplementary Figure 1: "Vegetation cover" I don't think sequencing data from middens are proper proxies for vegetation cover in the landscape. I would use a more tentative title like abundance

Corrected. Changed to 'plant abundance'

4. Supplementary Figure 2: a) Coverage plot: The green bar appears superfluous given the mean coverage is indicated on the graph; b)

Corrected. Green bars were removed

The meaning of arrows in the deamination graphs is unclear to me. The 5' arrow is pointing away from its prime end, while the 3' is pointing toward. The numbers on the X-axis are sufficient to indicate the directionality of the DNA 5' and 3' end.

Corrected. Arrows were removed

5. Supplementary Table 7. Column Protocol: They are both shotgun library protocols. The difference is with sampling type; helminth egg or sediment

Corrected. Changed to *Sample type: sediment/parasite eggs.*

Reviewer #3 (Remarks to the Author):

In their article entitled 'DNA evidence of bowhead whale exploitation by Greenland Paleo-Inuit 400 years ago', Seersholm et al. present new evidence for the early exploitation of bowhead whale within Paleo-Inuit sites in Greenland. Through the analysis of sedaDNA from midden deposits at four well-characterized archaeological sites, spanning multiple cultural occupations, they substantiate the range of taxa detected through previous zooarchaeological analyses, and detect previously un(der)represented species (bowhead whale, walrus, and caribou) as well as characterize plant communities at the site. The DNA analysis is completed to a high standard, and the authors follow published standards for the authentication of their data.

The authors assert that the detection of concentrated bowhead whale DNA within two Paleo-Inuit sites represents the most notable finding of the paper. Indeed the high proportion of bowhead whale DNA is intriguing, considering the lack of associated cetacean bone material, and does represent the first evidence of whale exploitation by Paleo-Inuit in Greenland. However, there is some question as to whether the exploitation of whales would be considered unexpected in terms of our understanding of the human behavior of coastal peoples. Whale bone is abundant in many Northwest

coast and Holarctic archaeological sites, and there is a long history of exploitation of cetacean resources worldwide dating back at least 10,000 years. More than 70 years ago Graham Clark (1947, Antiquity 21, 84-104) described the Neolithic use of whale in the economy of Northern Europe, noting zooarchaeological evidence for the widespread use of stranded whales, but also iconography supporting very early whale hunting (a publication which should be cited in Seersholm et al's paper).

We absolutely agree that Paleo-Inuit whale exploitation is not unexpected; However, there has - until the present study - been a lack of evidence because of the scarcity of cetacean bone remains in Paleo-Inuit middens, we have clarified this on lines 345-353 (see below).

We thank the reviewer for the reference to Graham Clarks study on whaling in prehistoric Europe, which we found very interesting. We have cited this particularly relevant paper on line 381.

It is also well established that cetaceans in particular are poorly represented in archaeological sites, due to the difficulty in transporting large carcasses from shore to a habitation site (as discussed in Smith and Kinahan 1984, The invisible whale, World Archaeology, 16, 89-97 and Szabo 2008, Monstrous Fishes and the mead-dark sea, and touched upon by the authors). Likewise, ethnographic evidence (predominantly from the Northwest Coast of North America) suggests a focus on blubber exploitation (with blubber from drift whales being highly prized), and to a lesser extent on meat, and with almost no evidence for the exploitation or use of whale bones (e.g., Monks 2001, Int. J. Osteoarchaeol. 11: 136-149) Therefore, the presence of whale sedaDNA but the absence of whale bone within the middens is interesting, but perhaps unsurprising in terms of our knowledge of coastal resource exploitation.

We completely agree with the reviewer. We have elaborated on this matter on lines 345-353, and included the references suggested.

“The underrepresentation of whale bones in archaeological sites is a well-known phenomenon, typically ascribed to difficulties in transporting large carcasses from shore to the settlement^{10,11} in combination with the higher value of blubber or meat compared to bones¹². In the arctic, several studies have suggested that the fossil record may underestimate the importance of whales to ancient Arctic cultures^{13,5,14}, however, the lack of suitable methods to detect remains of tissue like blubber and meat in sediment have prevented further investigations on this matter. As such, our findings represent the first tangible evidence that bone counts alone may underestimate large whales in Arctic midden remains.” p.7 l.345-353

Moreover, the scale of exploitation is not conclusively demonstrated by the DNA evidence. The authors note that for harp and fur seal there are good correlations between bone biomass and the proportion of DNA reads (where sedaDNA could be argued to be principally derived from the degradation of bone and relatively smaller

quantities of discarded meat, fat, skin, etc.,). However, extrapolating this relationship to large cetaceans is problematic, since the sedaDNA from whale is obviously derived almost exclusively from non-bone products (which may more readily leach host DNA into the soil). Ethnographic evidence suggests that whale blubber (and skin) was often boiled to render the oil, which was skimmed off. Discarding the waste water in the midden might account for a high concentration of whale sedaDNA, and perhaps even a high degree of DNA leaching through the midden. Interestingly, the authors found that the calculated biomass of domestic animals at Sandnes was three times higher than that expected from osteological evidence, noting that feces and urine can concentrate DNA in the archaeological record. This result reinforces the difficulties of resolving 'actual' biomass used, based on sedaDNA results. To strengthen the manuscript, the authors could be far more critical of their argument concerning expected relationships between bone biomass and sedaDNA concentrations.

The reviewer raises some very important points concerning whether or not it is possible to infer biomass from DNA concentrations. We have explored this further in 'Supplementary Discussion 3 – Inferring biomass from DNA profiles' and on lines 359-365 in the MS:

"In summary, our results demonstrate that large mammals such as caribou, walrus, narwhal and bowhead whale is underrepresented in the osteological record while domestic species such as cow, sheep, goat and dog is overrepresented in the DNA profile. Hence, in order to confidently reconstruct subsistence practices from midden remains, it is strongly encouraged to apply a combination of sedaDNA and morphological analyses, as both of these approaches may be misinterpreted when standing alone (discussed further in Supplementary Discussion 3)." p.7, l. 359-365

"Supplementary Discussion 3 – Inferring biomass from DNA profiles

The sedaDNA approach applied here provides an excellent means to investigate the taxonomic distribution across a variety of taxa based on a few grams of sediment. As demonstrated in Figure 3, there is a good correlation between the DNA read counts and the expected biomass for harp seal, ringed seal, birds and fox at Qeqertasussuk. However, as discussed in the manuscript, the DNA distribution might not always reflect the biomass of the different species, as, e.g. defecation and urine might inflate the DNA record for domesticated species. Hence, when analysing sedaDNA results, the DNA sources should be carefully considered and, if available, the DNA data should be correlated with osteological evidence. In this study, the identification of hardened blubber oil within the sediment at Qeqertasussuk (Morten Meldgaard, personal communication) together with the absence of associated cetacean bones, suggests that the main source of bowhead whale DNA at Qeqertasussuk is blubber and meat. Alternative sources of DNA from marine mammals in this study could arise from the processing and usage of blubber. Blubber from seals or whales were used as fuel in lamps¹⁵; If such lamps were emptied onto the midden, the DNA signal from marine mammals could be inflated. Similarly, the

wastewater from boiling of skin and blubber in order to retrieve oil could have been discarded at the midden. However, the contribution of such alternative sources of DNA is unlikely to be significant as the blubber was heated, causing the DNA to be heavily damaged.

In summary, based on the presented evidence, it cannot be conclusively shown that the biomass for bowhead whale can be inferred directly from the DNA read counts, as is the case for harp seal. However, it can safely be concluded that the level of bowhead whale exploitation at Qerqertasussuk and Qajaa, by far exceeds what has been estimated from the bone record previously.” SI p. 13-14

A high proportion of whale sedaDNA is also not strong evidence for whale hunting as opposed to whale scavenging. Archaeological investigations at sites which span time periods involving the shift from scavenging to active hunting show limited differences in terms of the quantity of whale bone at a site, even through active whale hunting is thought to result in a greater quantity of whale biomass (e.g. Monks 2001, Arctic Anthropology, 38, 60-81). Although the authors note that the observed sedaDNA results may be a result of scavenging draft carcasses, there is a focus on whale hunting in the manuscript, even though the former explanation is the most parsimonious. More discussion on the widespread exploitation of stranded cetacean carcasses (observed in virtually all coastal economies where cetaceans strand and which continues even after the development of active hunting) would provide a more balanced interpretation of the results.

We agree with the reviewer on the point that high DNA concentration is not in itself evidence of whale hunting as opposed to scavenging. To provide a more balanced discussion, we have expanded the paragraph on whale scavenging:

“As opposed to whale hunting, scavenging of stranded cetacean carcasses was common in pre-historic times and has been described across multiple sites in Europe¹⁶, North America¹⁷ and Africa¹⁰. Hence, the relative abundance of bowhead whale DNA in the Saqqaq sediment layers could be explained by scavenging whale carcasses. Whales were probably abundant along the nutrient-rich West Greenland waters that were so attractive to European whalers, and dead (“drift”) whales could have been driven ashore by wind and tides making them available to Saqqaq beach-comers. The warmer Saqqaq climate may also have influenced the frequency of whale strandings; Today killer whales appearing in the less ice-congested Arctic waters often cause whales and other sea mammals to seek shelter in shallow bays and inlets, causing them to strand. Dependent on the rate of decomposition, the meat and blubber from drift whales might have been used for human food, oil lamps or feeding of dogs.” p.7-8, 1.380-416.

It is perplexing that although complete mtDNA genomes of bowhead whale were constructed to a relatively high depth of coverage, the authors did not attempt to

distinguish the minimum number of individuals that might be represented by the DNA sequences. Based on the data currently provided in the paper, for both middens it appears that the DNA sequences derive from only a single whale, respectively. A finding of several, distinct haplotypes present within the sedaDNA (e.g., through mtDNA genome capture), would provide stronger evidence for the continued and extensive exploitation of whale products.

This is a very relevant question that we have been looking into previously. We agree with the reviewer that it is very likely that the DNA presented in this study originates from several individuals. However, because the bowhead whale mitochondrial genomes have low coverage, it is difficult to distinguish potential haplotypes from sequencing errors. This renders an MNI analysis of the bowhead whale DNA equivocal. To illustrate this, we have included our results on this matter below, based on the 2 harp seal mitochondrial genomes and the 2 bowhead whale mitochondrial genomes:

To distinguish different individuals in the data, positions with less than 100% consensus support were investigated. In order to avoid a potential bias from ancient DNA damage, the following bases were ignored in the analysis: Thymidine bases within the first ten base pairs of a read where the consensus base was Cytosine and Adenine bases within the last ten bases of a read where the consensus base was Guanine. Furthermore, to account for sequencing errors, haplotypes represented by a single base at one position were ignored.

sample	coverage	SNPs	mean 1st	mean 2nd
QA6_39089	24.1	113	80.5%	19.5%
QT3_39089	3.5	9	68.9%	31%
QA6_27602	1.3	1	50%	50%
QT3_27602	3.5	14	71.6%	28.4%

Table 1. Positions identified with less than 100% consensus support (SNPs), when accounting for DNA damage and sequencing errors. Mean 1st and mean 2nd represents the mean abundance of the most abundant (1st) and the rare (2nd) base on each position. 39089: *Pagophilus groenlandicus*, 27602: *Balaena mysticetus*.

As depicted in Table 1, between 1 and 113 positions were identified across the four genomes with consensus support below 100% when accounting for DNA damage and DNA sequencing errors. This could suggest an MNI of 2 for all of these genomes. Moreover, for the harp seal genomes at Qajaa and QT, 7 and 1 pairs, respectively, of such positions were sufficiently close that both positions were covered by several reads. As presented in Table 2, three different haplotypes were identified at these positions. This may imply that the two harp seal mitochondrial genomes are represented by at least three individuals; however, this is based on unacceptably low counts of each haplotype.

positions	1st	2nd	3rd
QA6_39089			
1370+1371	GA:46	GG:2	AA:2
4354+4363	TT:10	CC:4	CT:4
4396+4405	AG:26	GG:4	AA:2
7450+7453	AC:23	GC:2	AT:2
12948+12950	CG:13	CA:5	TG:3
13306+13312	GA:24	AA:3	GG:2
14844+14859	TG:5	CA:3	TA:2
QT3_39089			
6747+6750	AC:5	GC:2	AT:2

Table 2. Pairs of potential SNPs within less than 100bp distance from each other. 39089: *Pagophilus groenlandicus*, 27602: *Balaena mysticetus*.

In conclusion, we believe that the harp seal and bowhead whale DNA presented in this study originates from several individuals, however we are reluctant to include the above results in the manuscript, as they are backed up by a very small data set. We agree with the reviewer that the way to approach this question would be to perform a whole mitogenome capture of the bowhead whale DNA. However this is beyond the scope of this paper.

To clarify that the mitochondrial genomes could be based on DNA from several individuals, we have included a short paragraph addressing this:

“While these mitogenomes serve as evidence for the identification of harp seal, bowhead whale and Taenia hydatigena, they should not be regarded as sequences from a single individual. Rather, these mitogenomes most likely represents a subset of the individuals present in the sediment layer from which they were retrieved.” p.5, l. 263-267

Following from that point, although I wholeheartedly agree with the authors that historians and archaeologists frequently underestimate the maritime technologies and capabilities of native cultures, the evidence provided in this paper can only provide the first concrete evidence for the early exploitation of whale products by the Paleo-Inuit, but is not particularly convincing of active hunting. The latter conclusion would need to be supported by far more evidence for the exploitation of multiple whale individuals that surpassed expectations for bowhead whale strandings in the area.

We agree that it is very difficult to discern scavenging from hunting based on the evidence provided on this study. We hope that the current version of the manuscript provides a balanced discussion on the means by which the whale product could have been obtained (lines 367-431).

Although the analysis of sedaDNA to document the use of animals within archaeological sites or palaeosols is now well established (particularly within arctic or Holarctic contexts), the use of shotgun sequencing to detect host-specific parasites is a novel approach for documenting taxonomic diversity within archaeological sites, and an interesting complement to zooarchaeological and sedaDNA approaches.

In sum, this is an interesting paper, with robust and novel sedaDNA data important for understanding animal exploitation and economies in Greenland. At the moment, too much emphasis is placed on the 'notable' detection of whale sedaDNA, which undermines some of important data on other identified taxa (including plants), and the novel use of parasite sedaDNA to understand species distributions. This paper makes an important contribution to archaeological understandings of subsistence economies within Greenland, but would perhaps be more appropriate for a specialized journal in zooarchaeology or Holarctic archaeology.

REVIEWERS' COMMENTS:

Reviewer #1 (Remarks to the Author):

The authors have done an excellent job at detailing their modified responses to all reviews. I believe this paper is a very nice addition now to the sedimentary aDNA literature and look forward to seeing it in print.

While I see the benefits of supporting the data in the ms (whale and seal exploitation) with as much additional circumstantial evidence, I still find the plant data of little value here. I think moving them to the SOM makes most sense - but am not married to this position.

Great ms.

Reviewer #2 (Remarks to the Author):

The authors have successfully addressed all my comments in the revised manuscript. I recommend the manuscript for publication without any further changes.

Reviewer #3 (Remarks to the Author):

I am pleased to re-review Seersholm et al.'s article entitled 'DNA evidence of bowhead whale exploitation by Greenland Paleo-Inuit 400 years ago'. Their revised manuscript included a more balanced review of scavenging vs hunting of cetacean remains, as well as a more expansive discussion of methods and issues in equating bone biomass with sedaDNA results. I also appreciated that the authors included in their rebuttal a discussion of the mtDNA genome analysis and the difficulty in resolving MNI counts, which I agree is problematic without deeper coverage of the mtDNA genomes. I believe that the authors have addressed all the issues raised in my initial review.

Note: There are two very minor grammatical errors in the revised version (lines 310-311, and 359-361).